# Nuisance Algae in Ballast Water Facing International Conventions. Insights from DNA Metabarcoding in Ships Arriving in Bay of Biscay

**Alba Ardura** [1],*, **Yaisel J. Borrell** [1] , **Sara Fernández** [1,2] , **Mónica González Arenales** [3], **José L. Martínez** [4] **and Eva Garcia-Vazquez** [1]

1   Department of Functional Biology and 4: DNA Sequencing Unit, University of Oviedo, 33006 Oviedo, Spain; borrellyaisel@uniovi.es (Y.J.B.); sara.fernandezfernandez@gmit.ie (S.F.); egv@uniovi.es (E.G.-V.)
2   Marine and Freshwater Research Centre, School of Science and Computing, Galway-Mayo Institute of Technology, Dublin Road, H91 T8NW Galway, Ireland
3   Sustainability Department. Port of Gijon. El Musel s/n, 33212 Gijon, Spain; mgarenales@puertogijon.es
4   Sequencing Unit, University of Oviedo, 33006 Oviedo, Spain; jlmf@uniovi.es
*   Correspondence: arduraalba@uniovi.es

**Abstract:** Ballast water is one of the main vectors of transport of nuisance species among marine ports. Neither treatment nor interchange completely reduces the risk of ballast water containing DNA from harmful species, being a signal of potential threat. However, although there are some efficient treatments, they are not available on all ships and there might be some technological/economical constrains for their active and routine usage. Understanding what routes lead to a higher risk of contamination is important for designing targeted surveillance. We analysed ballast water from seven ships arriving in Gijon port (south Bay of Biscay, Spain). DNA metabarcoding was employed for identification of exotic species and harmful algae. One ship carried DNA of 20 risk species in the ballast water. Three ships contained DNA of only one risk species, and three ships had none. Seventy two algae species were found, 22.2% are exotic to the Bay of Biscay and 11.1% are catalogued as harmful. The results demonstrated the importance of continuous surveillance of ballast water.

**Keywords:** ballast water; nuisance species; eDNA metabarcoding; harmful algal blooms (HABs); port routes

## 1. Introduction

Ballast water (BW) is used in ships for maintaining balance and preventing instability and accidents when travelling unloaded. Although it is necessary for the normal functioning of maritime transport, it has been the focus of increasing attention and has generated alarm in environmental forums when recognized as a vector of harmful species. BW may contain a variety of potential nuisance organisms: From lethal bacteria such as some strains of *Vibrio colera* that caused cholera epidemics in the 90s [1] when epidemic cholera was reported in Peru and rapidly spread through Latin America and Mexico—in July of 1992, *Vibrio cholerae* was found in the USA and the Food and Drug Administration (FDA) determined that it came from ballast water of ships whose last port of call was in South America [2]. To invasive aquatic species—IAS [3], and harmful algae responsible for toxic algal blooms [4,5]. The international community became aware of this problem and established rules and protocols to prevent the occurrence of new biological accidents caused by BW. The signature and entry into force of the International Ballast Water Management Convention (IBWMC) on 8 September 2017 was the main advance. Ratified from 63 Contracting Parties (countries) that represent more than

68% of world merchant shipping tonnage, the IBWMC sets strict limits to the number of plankton individuals and microbes in BW discharges.

Harmful algae, or HABs (harmful algal bloom producers), are algae that experience growth outbreaks and dominate the communities when the environmental conditions are favourable. Many of them have toxins that cause huge impacts in aquaculture and fisheries. Their outbreaks are known as red tides. It has been proposed that BW was the vector responsible for transporting HABs (van der Berg et al. [6]), and studies from ports (e.g., Butrón et al. [5]) and BW (e.g., Lilly et al. [4]), Pertola et al. [7]) confirmed it. In the IBWMC the limit to the plankton size range of 10–50 μm is <10 viable organisms per ml. This fraction contains practically all HABs, since microalgae cells are typically small, in the range of micrometers to tens of micrometers [8].

Dealing with BW is a challenge [9]. First, the complex biotic community transported in BW is difficult to manage because animals, algae, and bacteria may respond differently to treatments. For example, BW treatments based on water interchange reduce risk of zooplankton non-indigenous species (NIS) introductions across ecosystem types but are less effective with phytoplankton [10]. Second, the control methods employed to check the compliance of IBWMC requirements still need improvements. So far there is no single method available for rapid and easy monitoring of the whole biota (Eukaryotes and Prokaryotes) from ballast water or any water. Current methodologies rely upon water filtering, visual observation, sorting, and taxonomic identification of individuals from different water volumes depending on the plankton fractions. While it is true that there are some significant advantages of the microscopy-based identification of the biota in BW, such as the determination of the relative abundances of individual taxa or avoiding the phenomenon of the Polymerase Chain Reaction (PCR)-bias in metabarcoding methodology. Visual observation requires some time and the intervention of experts, and it is not possible to check if the strict requirements of waterborne organisms are met or not in a reasonable time of less than a few days.

Improvements in BW monitoring and treatment must be implemented to minimise the spread of biological invasions worldwide. Amongst others, there is an urgent need for identifying risk species and risk donor ports for being able to adopt tailored prevention measures.

Rey et al. [11] proposed DNA metabarcoding, based on high throughput sequencing (also called next-generation sequencing, NGS), as an approach of choice for monitoring BW, to detect DNA from harmful and in general undesired species, being a signal of potential threat. In this study, we will use this methodology that has been already validated for en-route BW studies (e.g., Ardura et al. [12], Zaiko et al. [13,14]), using ribulose-1,5-bisphosphate carboxylase/oxygenase gene, rbcL, as a metabarcode. This is a marker of choice for genetic inventories of phytoplankton [15] and has been employed in BW [13]. The mitochondrial DNA cytochrome oxidase subunit I (mtDNA COI) molecular marker has been used as a second marker. Our study aims at building upon the current role of BW as a carrier of potential nuisance algae, both HABs and NIS, using NGS metabarcoding. In this proof of concept, we use the next-generation sequencing (NGS) methodology to investigate the BW-mediated nuisance algae introduced to the southwest Bay of Biscay comparing BW arriving at the main international port of the region (Gijon) and port waters.

## 2. Materials and Methods

### 2.1. Samples Analysed

The region studied in this work is Asturias (southwest Bay of Biscay, Spain), located at 43°33′23″ N, 5°41′57.98″ W, centred in Gijon which is the main international port of the region. Gijon port has a temperate, rainy oceanic climate within the Atlantic Arc. It received 16,787,945 and exported 3,357,136 tons of cargo in 2017.

Two sets of NGS samples were considered in this study: BW samples from ships arriving in Gijon port in 2017 (Table 1); and water samples from Gijon's and neighbouring ports in 2015. Data from Gijon's and surrounding were already studied and published in Borrell et al. [16]. BW samples will be

compared to previously published data from the surrounding phytoplankton habitats. The locations of origin of the samples analysed, i.e., where the BW was uploaded, and the port of Gijon, are shown in Figure 1.

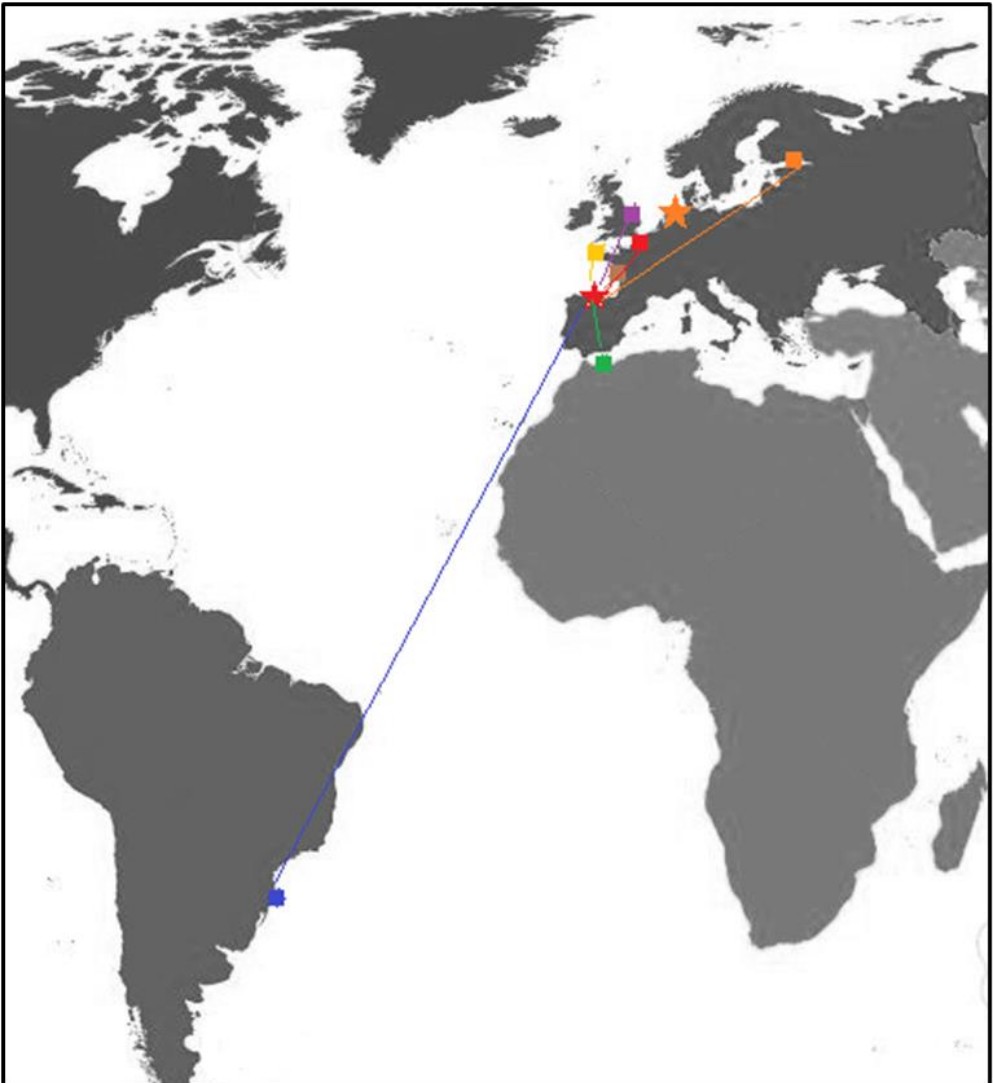

**Figure 1.** Map showing the ports where the ballast water (BW) was uploaded in the ships analysed. Port of Gijon and Bremerhaven port are marked with a red and orange star, respectively. The ports marked with squares are the ports of origin of the ships arriving in port of Gijon analysed in this study (yellow: Brest, green: Nador, blue: Tubarao, purple: Gunnes Warf, brown: La Pallice-La Rochelle, orange: Vystosk, red: Rotterdam).

**Table 1.** Ballast water analysed, number of algae species, and number of risk species (exotic and/or harmful) detected employing DNA metabarcoding methodology in this study.

| Ship | Type of Ship | Origin Port | Country | Region | Sampling Date | Days since BW Upload | Species Detected | Risk Species |
|---|---|---|---|---|---|---|---|---|
| Ship#1 | General cargo | Brest | France | English Channel/Atlantic Ocean | 31/08/2017 | 3 | 1 | 0 |
| Ship#2 | LPG tanker | Nador | Morocco | Mediterranean Sea | 09/10/2017 | 39 | 4 | 1 |
| Ship#3 | Bulk carrier | Port of Tubarao | Brazil | SW Atlantic Ocean | 19/10/2017 | 3 | 1 | 1 |
| Ship#4 | General cargo | Gunnes Warf | United Kingdom | Fluvial port (River Trent), North Sea | 12/12/2017 | 12 | 66 | 20 |
| Ship#5 | Oil tanker | La Pallice (La Rochelle) | France | NW Bay of Biscay/Atlantic Ocean | 05/07/2017 | 1 | 1 | 1 |
| Ship#6 | Bulk carrier | Vysotsk | Russia | Baltic Sea | 11/07/2017 | 4 | 0 | 0 |
| Ship#7 | General cargo | Rotterdam | Netherlands | Fluvial port, North Sea | 13/07/17 | 1 | 0 | 0 |

The BW samples were taken from ships arriving in Gijon port in the last half of 2017, just after the entry into force of the IBWMC. The BW control in the origin port being not compulsory in Spain then, the captains should give their approval for BW sampling. Port authorities asked the captains of all the ships with BW arriving in Gijon port between July and November of 2017 to allow port staff on board to take a sample. The captains of seven out of 38 requested ships (18.4%) accepted that a sample of BW was taken from the ship they governed. Two litres of BW water were manually pumped through the sounding pipe and stored refrigerated (4–8 °C) until laboratory analysis.

Environmental DNA was extracted from filters (one extraction per filter) with the PowerWater® DNA Isolation Kit (MoBio Laboratories, Carlsbad, CA, USA) under controlled airflow conditions using a laminar flow PCR hood. The extraction followed the manufacturer's instructions. In total, eight extractions: Seven extractions (one per ship) and one filtering negative control, were made. An extraction negative control was included at this step to monitor possible cross contamination. The process followed with port water samples was thoroughly described in Borrell et al. [16]. Metabarcoding based on NGS was used.

## 2.2. DNA Analysis

The extracted DNA from 2017 filters was sent to Macrogen (Seoul, Korea) for further analysis. The DNA was quantified by a fluorescence-based method Victor 3 (Picogreen, Invitrogen). The modified universal COI primers mlCOIintF [17] and jgHCO2198 [18] were used for polymerase chain reaction (PCR) amplification of a fragment of 313 base pairs (bp) within the mitochondrial gene coding for the cytochrome oxidase subunit I (COI). A fragment of 312 bp within the plastid rbcL gene was PCR amplified using Diat-rbcl-708F and Diat-rbcL-R3 designed for diatom identification [19]. The primers (Supplementary Table S3) were modified to include Illumina™ overhang adaptors and sample-specific indices following the dual-PCR Illumina protocol (https://support.illumina.com/), where the conditions for the amplicon PCR were changed to the ones described by Leray et al. [17] and Rivera et al. [19] for COI and rbcl genes amplification, respectively. In addition, bovine serum albumin (BSA) was added to the PCR reactions to increase PCR yields from low purity templates and to avoid, as much as possible, the effect of inhibitors presents in the water. After library construction, MiSeq Illumina platform was employed to run the sequencing step using paired-end sequencing ($2 \times 301$). Adapters and indices were removed from the raw data along with reads <36 bp using the Scythe and Buffalo software, respectively.

## 2.3. Bioinformatics and Statistical Analysis

For the NGS obtained from Gijon port ships, in the primary quality filter Illumina package bcl2fastq v1.8.4 and Fastq Quality Encoding: Sanger Quality (ASCII (American Standard Code for Information Interchange) Character Code = Phred Quality Value + 33) were employed. The trimmed data in FASTAQ format (a text-based format for storing both a biological sequence (usually nucleotide sequencing) and its corresponding quality scores) were further processed using the next-generation microbiome bioinformatics platform QIIME [20]. The paired-end reads from each sample were merged when they presented an overlapping region of at least 100 bp and less than 15% of differences within this region. All the sequence reads were assessed for quality by applying a Phred quality score threshold of 20 and were filtered by length (200 bp ≤ reads ≤ 400 bp) from the downstream analysis.

For the taxonomic assignation, BLAST alignment was compared with the NCBI database (https://www.ncbi.nlm.nih.gov/) of COI and rbcL sequences (obtained in September 2017 and May 2018, respectively) used as threshold criteria: Maximum E-value = $1 \times 10^{-50}$ and minimum percent identity = 80.0 for COI and 85.0 for rbcL, enough for genre assignation. Finally, operational taxonomic units (OTUs) tables, for each sample and the number of sequences assigned to them, were constructed clustering reads with a 100% identity between them and maintaining all sequences including singletons to retain maximum sensitivity for species detection. The reason is that, although the removal of singletons is usually employed to eliminate false positives [21], a false negative could be costly in the

context of species survival, early detection of NIS or marine biosecurity surveillance [22]. Sequences of organisms without relevance to the study (e.g., human, insects, plants, etc.) were removed from the dataset.

The taxonomic information and native/non-native distribution from the remaining OTUs was checked against the World Register of Marine Species (http://www.marinespecies.org/) and AlgaeBase (http://www.algaebase.org/). HAB species were determined based on the previous published studies.

The methodological description of bioinformatic analysis done for port water samples was thoroughly described in Borrell et al. [16].

Comparisons of taxonomic groups distributions among samples were done using the contingency chi-square analysis.

## 3. Results

### 3.1. Algae Community in BW of Ships Arriving in Gijon Port

The BW of the seven ships sampled was analysed and provided very different results depending on the ship, apparently without any relation with the ship's origin and time since the BW was uploaded (Table 1). The NGS results of DNA concentration, number of reads before and after quality filters, and number of reads assigned to an OTU are presented in Table 2. All the samples provided DNA and sequence reads from the two metabarcodes, but differences between replicates was found for most ships with a COI barcode and for Ship#4 with a RBCL barcode. After the first quality filter with the Illumina package a total of 273,270 and 201,956 reads were obtained for COI and rbcL, respectively. From the QIIME pipeline a total of 25,925 and 10,609 reads successfully assigned taxonomically from which 17,356 and 9372, respectively corresponded to marine taxa (Table 2). Combining the results of the two genes, a total of 72 algae species were found, most of them in Ship#4 (Supplementary Table S1—fastaq files available in https://www.ncbi.nlm.nih.gov/bioproject/PRJNA604698). From them 16 (22.2%) are exotic to the Bay of Biscay and eight (11.1%) are catalogued as harmful for their bloom forming capacity, toxicity, or both. Therefore, in total 24 species (33.3%) could be considered potential nuisance algae.

**Table 2.** Next-generation sequencing (NGS) results obtained from the ballast water samples analysed, presented as the number of reads obtained per sample, assigned taxonomically from QIIME pipeline, and from marine taxa (after expert check). QC: Quality control. COI and rbcL are cytochrome oxidase I and ribulose-1,5-bisphosphate carboxylase/oxygenase gene metabarcodes, respectively.

| | | COI | | | rbcL | | |
|---|---|---|---|---|---|---|---|
| **Ship** | **Sample** | **QC Reads** | **Assigned** | **After Expert Check** | **QC Reads** | **Assigned** | **After Expert Check** |
| Ship#1 | 1 | 130 | 0 | 0 | 538 | 0 | 0 |
| | 2 | 26 | 2 | 0 | 244 | 1 | 1 |
| Ship#2 | 1 | 138,174 | 9.234 | 3.027 | 73,400 | 372 | 372 |
| | 2 | 6 | 0 | 0 | 72,450 | 0 | 0 |
| Ship#3 | 1 | 634 | 98 | 47 | 14 | 0 | 0 |
| | 2 | 24 | 0 | 0 | 18 | 1 | 1 |
| Ship#4 | 1 | 131,180 | 16.585 | 14.281 | 50,044 | 10.234 | 8.997 |
| | 2 | 4 | 0 | 0 | 62 | 0 | 0 |
| Ship#5 | 1 | 32 | 0 | 0 | 1280 | 1 | 1 |
| | 2 | 80 | 1 | 0 | 1160 | 0 | 0 |
| Ship#6 | 1 | 24 | 0 | 0 | 1694 | 0 | 0 |
| | 2 | 22 | 0 | 0 | 1020 | 0 | 0 |
| Ship#7 | 1 | 28 | 1 | 0 | 22 | 0 | 0 |
| | 2 | 2906 | 4 | 1 | 10 | 0 | 0 |

Analysing the results individually by ship, Ship#6 and Ship#7 did not provide any significant assignment with algae species. In the other four ships less than five species were identified (Table 1, Supplementary Table S1): One exotic in Ship#1 from Brest (*Aulacoseira granulata*), one exotic in Ship#3 from Tubarao (*Amphora pediculus*), one native from European waters in Ship#5 from La Rochelle (*Cyclotella meneghiniana*), four species (from which one harmful, *Heterosigma akasiwo*) was in Ship#2 from Nador. Finally, 66 species were found in Ship#4 from which 20 (30.3%) were potential nuisances (Table 1, Supplementary Table S1). The major part of algae detected are native species from the NW Atlantic Ocean, the Baltic, and the North Sea, following by exotic species coming from Brazil, Gulf of Mexico, Caribbean, Asia, and the Indian Ocean. Harmful species (HABs) correspond to 10% of the species detected (Figure 2; Supplementary Table S1).

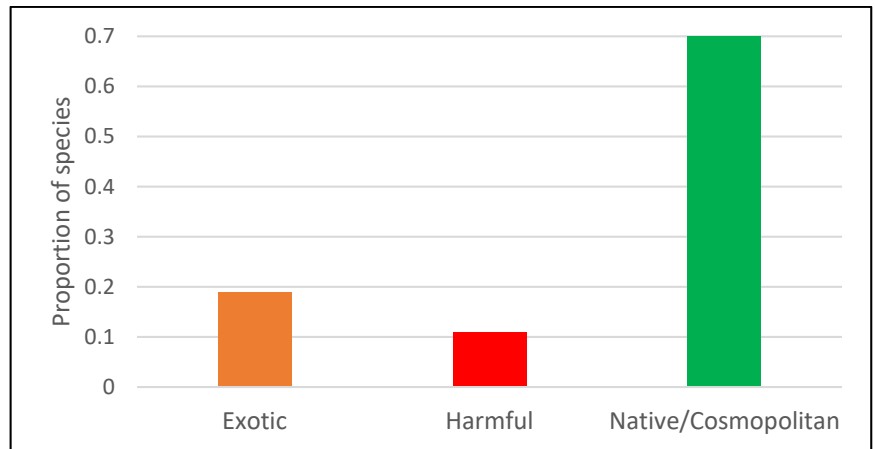

**Figure 2.** Proportion of algae species detected from eDNA in the ships analysed, for each type of alga considered (harmful algal bloom (HAB): Non-indigenous species (NIS), native from the NW Atlantic Ocean, the Baltic, and the North Sea).

Focusing on HABs (Table 3), six species were found in the ships studied: Two bloom-forming and four with toxicity. *Heterosigma akashiwo* and *Pseudo-nitzschia galaxiae* are exotic to European waters. *Chaetoceros* genus is considered an harmful algae due to the rigid (heavily silicified) nature of its setae which can irritate the gills of fish causing excessive mucous production or, in extreme cases, cause direct physical damage to the gills. The setae present in *Chaetoceros* genus contain barbs along their length exacerbating the amount of irritation and/or damage they can cause [23]. Although some species of *Chaetoceros* genus occur as native species of the phytoplankton all across European coasts, previous studies determined that phytoplankton blooms were dominated by this genus, being *C. septentrionalis* one of this species [24]. *Halamphora* species have mainly been demonstrated to be toxic from laboratory tests, they produced low concentrations of domoic acid [25]. However, two isolates of the same species were found to be nontoxic and there are doubts about the correct identification of this diatom owing to the difficulties in identifying correctly Halamphora species [26]. *Heterosigma akashiwo* has been associated with fish kill events within the aquaculture industry for many years. The precise toxicological mechanism involved in these fish kills is unclear; however, much research attention has focused on the production of reactive oxygen species (ROS), brevetoxin-like compound(s), excessive mucus, or hemolytic activity by these toxic algae ([27]; www.nwfsc.noaa.gov › efs › microbes › harmful_algae). *Pseudo-nitzschia galaxiae* produce domoic acid (DA) [28], which is a kainic acid-type neurotoxin that causes amnesic shellfish poisoning (ASP). Exposure to this compound affects the brain, causing seizures, and possibly death of shellfish.

**Table 3.** HAB species found from eDNA in the ballast water examined (presence marked with X).

| Species | Cause of Concern | Origin | |
|---|---|---|---|
| | | #2, Nador | #4, Trent |
| *Chaetoceros septentrionalis* | Harmful algae, HAB (23) | | X |
| *Blixaea quinquecornis* | Baltic, Bloom forming species (28) | | X |
| *Cylindrotheca closterium* | Cosmopolitan, Bloom forming (29) | | X |
| *Halamphora sp.* | HAB (4) | | X |
| *Heterosigma akashiwo* | NW Pacific, HAB (30) | X | |
| *Pseudo-nitzschia galaxiae* | NW Atlantic HAB (30) | | X |

*3.2. Comparison of BW and Port Phytoplankton Detected Through NGS in the Region*

The algae arriving in BW and the community detected using the NGS method from port waters in the region were quite different (Supplementary Table S2). Most comparisons are done at genus level because the taxonomic resolution of most OTUs could not reach the species level in the study of Borrell et al. [16]. Of the 86 algae species identified, 81.4% were Chromista and the rest were Plantae. Only eight were shared by BW and port waters: Eight Chromista and the green alga *Picochlorum* (Figure 3). Port waters contained more Plantae than BW (Figure 4A), the taxonomic profile being clearly distinct with more Bacillariophyceae in BW and more Chlorophyta classes in port water (Figure 4B). Of the shared species only one could be considered nuisance: The exotic *Nitzschia palea*, native to the Gulf of Mexico (Supplementary Table S2). Other exotic algae occurred in the port that were not found in these ships' BW. HABs were not identified from port waters in Borrell et al. [16] work because the taxonomic resolution at genus level is not enough, but some genera detected such as *Gyrodinium* and *Odontella* contain HABs and/or invasive species.

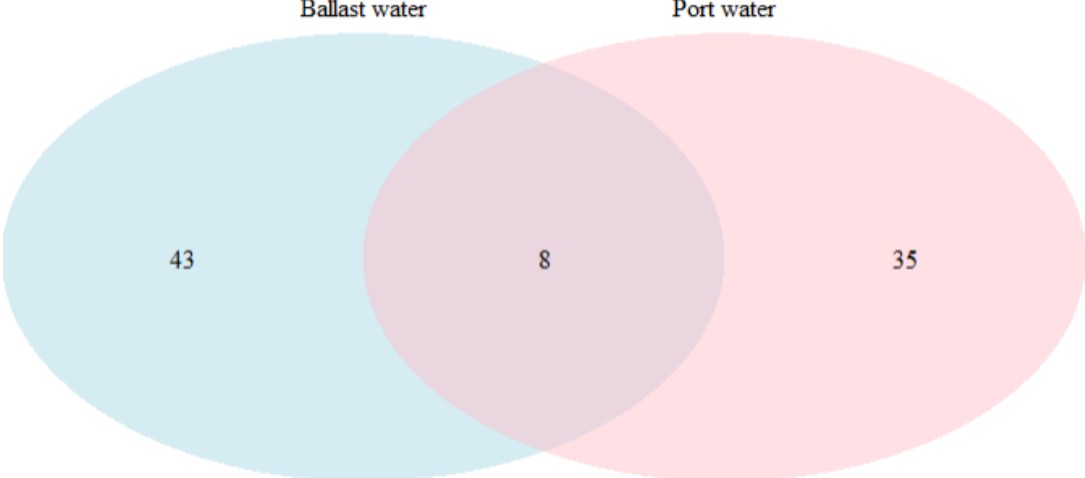

**Figure 3.** Venn diagram showing the number of unique species present in BW and port water and overlapped species between them.

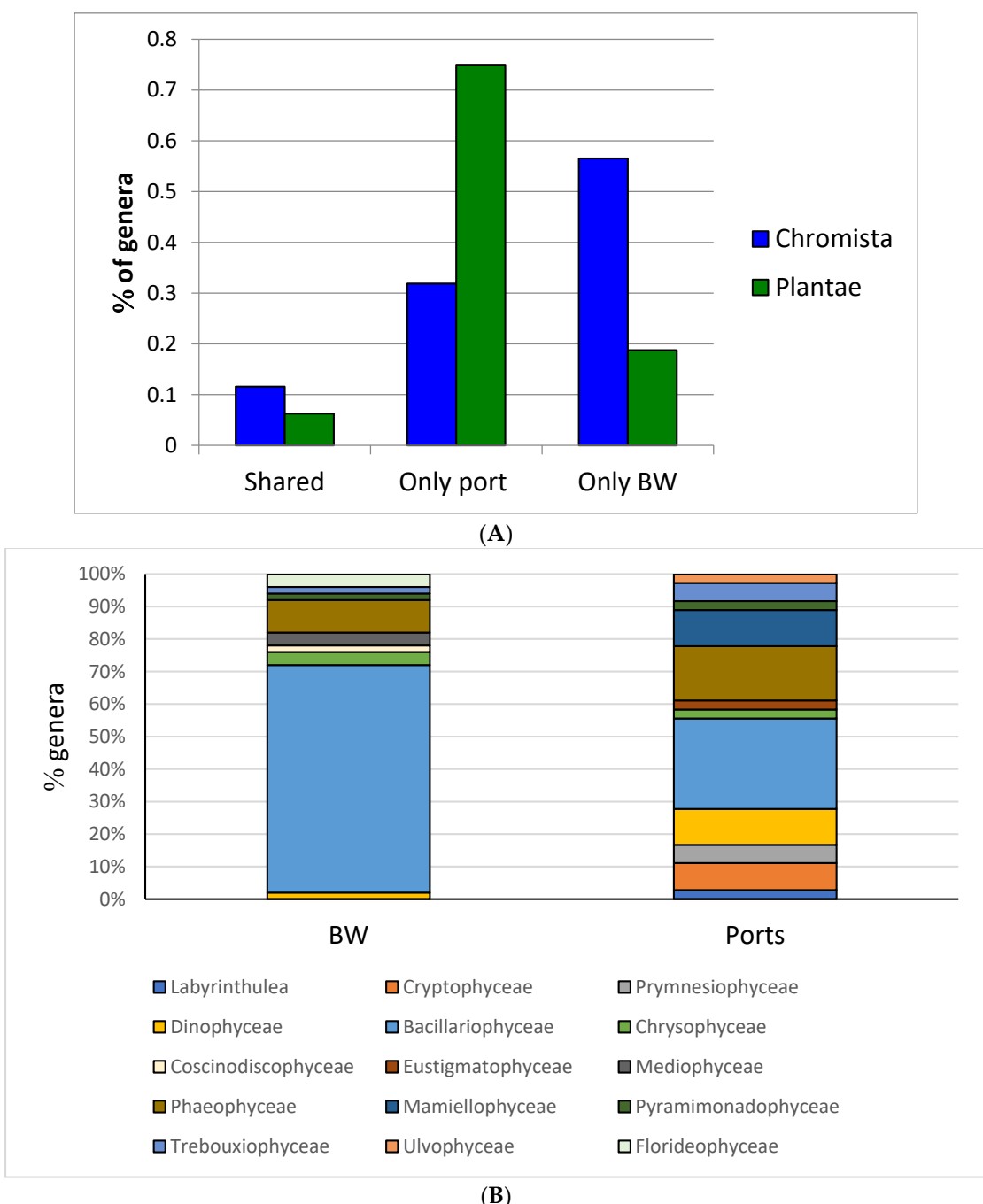

**Figure 4.** Comparison of algae found using NGS from ports and BW in Asturias region (SW Bay of Biscay). Results are presented as: (**A**) The proportion of genera of each type of alga (Chromista or Plantae) found only from ports, from BW, or shared between the two habitats; (**B**) taxonomic profile of BW and port water, as a proportion of genera by class of algae.

Further comparison of BW and port waters was done considering the geographic region from where nuisance algae are native (Figure 5). The difference in the distribution of nuisance algae by native region between BW and port waters was statistically significant (Chi Square = 12.6, 6 d.f., $p$ = 0.04; Monte Carlo $p$ = 0.037). While most nuisance algae found from metabarcoding in port waters were from the Baltic Sea (Figure 5), most BW species were from the Northwest Atlantic (North American), although none of the ships came from North American ports.

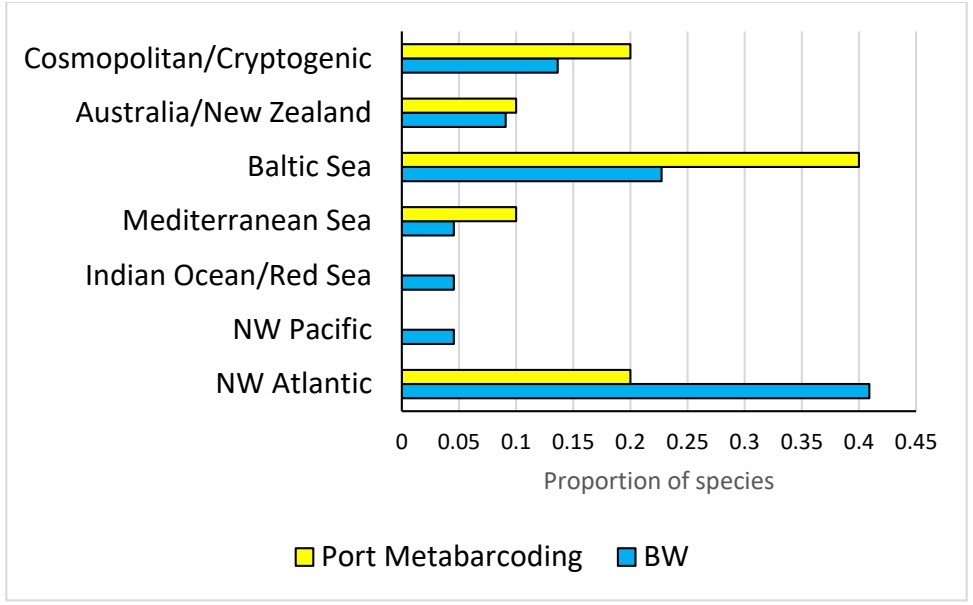

**Figure 5.** Proportion of non-native algae species from different regions found in BW released in Gijon port, compared with non-native algae detected in port waters from eDNA metabarcoding (Borrell et al. 2017).

## 4. Discussion and Conclusions

The results obtained in this study revealed several issues that are important for application to the control and management of ballast water. After the entry into force of the IBWMC we have detected some species of HABs that are dangerous from BW of ships entering Gijon port, using NGS metabarcoding. *Cylindrotheca closterium* has been found from BW in cold Baltic ports (6). *Heterosigma akashiwo* and *Pseudo-nitzschia galaxiae* were also found in ports of Southeast Bay of Biscay where they were probably introduced by BW [5]. The risk of them becoming established in Gijon port seems therefore to be real. Although they were not detected in previous metabarcoding surveys [16], they may be overlooked because in that study only a few litres of water were sampled from three locations within a port, thus the samples, although informative about the most abundant species, were not really representative of the port plankton. Moreover, it is important to remark that metabarcoding-based detection of a risky organism should be treated with caution, since although the detection of species DNA confirm the presence of the organism, it does not unambiguously confirm the presence of viable organism.

Technically, we can confirm the utility of NGS metabarcoding methodology for detection of nuisance species, thus for BW monitoring. Although the water samples taken from the ships arriving in Gijon port were very small and without any concentration, we were able to detect many algae species including some exotic and harmful algae. This does not mean the method is exhaustive and likely many species escape the control in such small BW volumes. Samples were taken from the sounding pipe, but it is known that the organisms travelling in BW are not the same in all the parts of the tank. In other words, the community inside the ballast tank is not spatially homogeneous, but patchy [29,30]. Sampling from the sounding pipe, as we did in our study, only retrieves a part, not higher than 60%, of the plankton that is retrieved from the tank manhole [31]. Even with these limitations, the method allows detection of at least part of the algae that are potential nuisances for the recipient ecosystem. However, the method is not completely mature yet and some improvements are recommended such as using RNA metabarcoding for surveying living individuals—RNA has a shorter life than DNA, creating standardized protocols of sampling and molecular analysis, and improving quantitative methods for expanding the surveys beyond detection [32]. Other molecular methods

targeting particular species, such as microarrays and quantitative PCR (qPCR) (e.g., Dittami et al. [24]), could complement metabarcoding approaches.

Focusing on zooplankton, Casas-Monroy et al. [10] concluded that ballast water exchange would reduce the invasion risks to freshwater but not so much to marine ecosystems because the environmental mismatch is lower between freshwater sources and marine recipients than the other way around. In the BW analysed we did find numerous algae that are freshwater species but apparently survived long travels. In the BW analysed here, the presence of many species described from North American freshwater suggests that it is probable that the conclusions from Casas-Monroy et al. [10] conclusions could be also applied for phytoplankton.

On the other hand, most species transported in BW were not detected from port waters. Although, as commented above, port water samples were not totally representative of port plankton, at least they reflected the most abundant species and contained DNA from some invasive invertebrates, amongst other species of interest [16]. One could say that the algae community travelling inside ballast tanks is very different from that found in the ports; in other words, most BW algae do not settle in the recipient waters. The waters of Gijon port are relatively clean (www.asturias.es). Clean ports with a rich native community seem to carry less non indigenous species (NIS) than ports with poor native communities [33], perhaps because they may have less niches available for newcomer species and this serves as a sort of invasion resistance [34]. Whatever the reason, the idea of a community of harmful algae travelling in BW still after the implementation of the IBWMC, with occasional releases in some ports, would be supported from our data and from data of port communities where the presence of HABs and invasive algae is common (e.g., Pertola et al. [7], Butrón et al. [5]).

**Supplementary Materials:** The following are available online at http://www.mdpi.com/2073-4441/12/8/2168/s1. Table S1: Algae species found from ballast water samples obtained from ships arriving in Gijon (SW Bay of Biscay) port in 2017 after the entry into force of the International Ballast Water Management Convention. Samples are coded as X.Y, X being the ship # and Y the sample. Presence 1, absence 0; Table S2: Algae found from ballast and port waters in Gijon region (southwest Bay of Biscay). Results are presented at the genus or species level depending on the taxonomic resolution reached in Borrell et al. (2017) where port water data come from. Algae shared in ballast and port waters are marked in bold; Table S3: Primer sequences for both COI and rbcl gene fragments.

**Author Contributions:** Conceptualization, A.A. and E.G.-V.; data curation, E.G.-V.; formal analysis, E.G.-V.; methodology, A.A. and S.F.; resources, Y.J.B. and M.G.A.; software, S.F. and J.L.M.; writing—original draft, A.A. and E.G.-V.; writing—review and editing, A.A., Y.J.B., S.F., M.G.A., J.L.M., and E.G.-V. All authors have read and agreed to the published version of the manuscript.

**Funding:** This research was funded by the Spanish Ministry of Economy and Competitiveness with the Grant BLUEPORTS MINECO CGL-2016-79209-R and FC-GRUPIN-IDI/2018/000201.

**Acknowledgments:** This study has been supported by the Spanish Ministry of Economy and Competitiveness with the grant BLUEPORTS MINECO CGL-2016-79209-R and FC-GRUPIN-IDI/2018/000201 and it is a contribution from the Marine Observatory of Asturias (OMA). We are grateful to the maritime police officers of Port of Gijon for their help with BW sampling. A.A. holds incorporation-Juan de la Cierva fellowship.

**Conflicts of Interest:** The authors declare no conflict of interest. The funders had no role in the design of the study; in the collection, analyses, or interpretation of data; in the writing of the manuscript, or in the decision to publish the results.

**Ethics statement:** This work was done in accordance with The Code of Ethics of the World Medical Association (Declaration of Helsinki) for animal use in research adopted by the 41st World Medical Assembly, Hong Kong, September 1989, revised by the 57th WMA General Assembly, Pilanesberg, South Africa, October 2006, and reaffirmed by the 203rd WMA Council Session, Buenos Aires, Argentina, April 2016. Anonymous treatment of the data was agreed with the authorities of Gijon port. They will be employed for research purposes only.

**Accessibility of molecular data:** The Fastaq files from the BW of ships arriving in Gijon port in 2017 are available in GenBank, reference PRJNA604698 (https://www.ncbi.nlm.nih.gov/bioproject/PRJNA604698). Port data is available in Borrell et al. (2017).

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
