# Peer review of "Nuisance Algae in Ballast Water Facing International Conventions. Insights from DNA Metabarcoding in Ships Arriving in Bay of Biscay"

_water, doi:10.3390/w12082168_

Round 1
Reviewer 1 Report
There are many small errors in grammar and English usage. I have corrected them in a Word.docx document

Author Response
Dear Reviewer,
thank you very much for your comments and your effort to correct the errors. All of them have been modified in the last version following your advices and the manuscript have been improved a lot.
Thanks,
Alba.
Reviewer 2 Report
This is potentially an interesting and useful study that should, eventually, probably be published in the journal. However, there are some obvious shortcomings that require substantial revision of the present manuscript. Most importantly, the primary data seem to unavailable for a review (see below the point no. 19). Besides that, a number of other points needs to be answered, supplemented or corrected.
1) l. 19-20: " just after the entry into force of the International Ballast Water Management Convention in 2017"
Is it reasonable to expect that the entering the Convention into effect might have had any real effects on actual composition of the ballast in the studied ships? If not, then, I think this shouldn't be included in the "Abstract".
2) Abstract - Shouldn't there by some mention on the total diversity of algae recovered from BW samples?
3) l. 32-33: "that caused cholera epidemics in the 90s"
Can you be more specific? In which countries did this happen?
4) l. 47: "This fraction contains practically all HABs."
But there surely must be some with cellular dimensions exceeding 50 micrometers. If this is the case, than it would be useful if you can mention these exceptions by name.
5) l. 48: "Therefore, dealing with BW is a challenge..."
The word "Therefore," at the beginning of the sentence is clearly redundant.
6) l. 51: NIS
This abbreviation has not been explained in the text.
7) l. 52+55: "–employed", "–from"
Delete "–".
8) l. 55-59: "Current methodologies rely upon water filtering, visual observation, sorting, and taxonomic identification of individuals from different water volumes depending on the plankton fractions. This requires some time and the intervention of experts, and it is not possible to check if the strict requirements of waterborne organisms are met or not in a reasonable time of less than a few days."
Well, yes, this is true, but this is also not the whole story. Of course, there are also some significant advantages of the microscopy-based identification of the biota in BW. Indirect methods (such as metabarcoding) inevitably result in distortion of the relative abundances of individual taxa, which can, on the other hand, be easily accounted for during microscopic observation. In addition, thewell.known phenomenon of the PCR-bias in metabarcoding methodology leads to inestimable ommission of certain species from the resulting lists. This is also prevented by microscopic analysis.
Summing up, in this part you should not just include the drawbacks of the microscopic identification but you should also honestly mention the strong suits of this methodology.
9) l. 66 and elsewhere: "Ardura et al. 2015a, Zaiko et al. 2015a,b)"
I thought that the references should be included as numbers in brackets. Am I wrong?
10) l. 68-73: OK, but what is the specific goal of the study? This has to be clearly specified. It would be best if you could state the null hypothesis that was evaluated. However, in a purely exploratory study, the general aim of the investigation will suffice.
11) l. 75-81: I would expect that this "Ethics statement" should better be included somewhere at the end of the study near Acknowledgements. Here, in M&M it looks rather weird as it does not contain any information about actual methodology of the study.
12) l. 84-86: "Of temperate, rainy oceanic climate within the Atlantic Arc, Gijon port received 16.787.945 and exported 3.357.136 tons of cargo in 2017."
How does the climate of Gijon relate to the size of the port? Please, split this sentence into two.
13) l. 88-89: "neighbouring ports in 2015 (Borrell et al. 2017)"
Here, you should clearly state that the environmental samples were already studied in that previous study and, thus, that the BW samples will be compared to previously published data from the surrounding phytoplankton habitats.
14) l. 103: "corresponded to ships" < "were taken from ships"
15) l. 118: "Seul" < "Seoul"
16) l. 124: "The primers were modified to include..."
It would be quite useful to include the primers in a supplementary table. The reader could then check the analyzed part of rbcL (COI) gene to see how variable part of the actual gene sequence it represents, etc.
17) l. 143-144. "minimum percent identity=80.0 for COI and 90.0 - 85.0 for rbcL"
Well, I am not sure if this is not too generous for a sensible identification. For example the rbcL gene sequence tends to be quite conservative in its large portions. Surely, the average distance of two related but well delimited species or even genera across the rbcL gene sequence would easily be much less than 85 or 90 %...
18) l. 164-165: "The NGS results of DNA concentration, number of reads before and after quality filters, and number of reads assigned to an OTU are presented in Table 3."
This must be some kind of a mistake as the actual Table 3 does not include any of these data.
19) l. 171-172: "Combining the results of the two genes, a total of 72 algae species were found (Supplementary Table 1)."
Fine, but in this table only the species names are included. Surely, what is needed are the actual barcode sequences of individual taxa. These are the actual primary data of your study and these must be made available. Both sampled regions are quite short and it would not be a problem to create a table (or two) with 72 rows containing the sequence data for each of the identified species. Only this would allo a reader (or a reviewer) to check for the quality of the data and corectness of actual identifications. Without that the study remains a "black box" and - I apologize but - this is not acceptable.
20 l. 171: "72 algae species were found"
... and most of them in the ship#4; this should be mentioned at this point, too
21) l. 192-193: "Harmful species (HABs) correspond to 10% of species detected. (Figure 2; Suppl. Table 1)."
What criteria were used to judge which species are members of the "HAB" group? I think this was never explained in M&M, was it? The same applies for the term "exotic species". Which authoritative taxonomic monographs or studies were used as the source of information for such assignment of individual taxa?
22) l. 200: "Chaetoceros genus is considered harmful algae..."
OK, but are you aware that tens (if not hundreds) of Chaetoceros spcies occur as native species of the phytoplankton all across European coasts? It may be that C. septentrionalis is considered as HAB but why this one and not the others?
23) l. 215 and below: "Comparison of BW and Port Phytoplankton Detected Through NGS in the Region"
The comparison at the level of kingdoms (!) - Plantae/Chromista - is very crude. Please, at least you should illustrate by some kind of Venn diagrams how many of the taxa (= genera) occured in the BW only, in the native phytoplankton community only and in both groups.
24) l. 239: "Metabarcoding"< "metabarcoding"
Author Response
Dear Reviewer,
Thank you for your comments. The manuscript has been modified following your advice, which has been greatly improved.
All comments have been answered in the attached file.
Regards,
Alba.

Round 2
Reviewer 2 Report
I was going through the revised text, as well as through the authors' responses to my original review. In most points, the authors followed closely my remarks and revised the manuscript accordingly. A few things were explained or argued sufficiently so that I don't feel that it is now necessary to request additional revisions.
My only point of uncertainty remains the style of references in the text. As far as I have seen the other papers in this journal refer to individual citations by numbers in brackets (i.e. [1]). However, you use the full names and publication dates. So, this is up to the editor if she considers it to be acceptable or it needs to be corrected.
Author Response
Thank you very much, the references style has been modified.